# Depression and associated factors among older adults in Bahir Dar city administration, Northwest Ethiopia, 2020: Cross-sectional study

**Tamrat Anbesaw**  *, **Betelhem Fekadu**

Department of Psychiatry, College of Medicine and Health Science, Wollo University, Dessie, Ethiopia

* tamratanbesaw@gmail.com

## Abstract

### Background

Depression is the most common psychiatric condition among older adults, and it goes unnoticed by individuals themselves and is under-diagnosed by clinicians due to the misconception that these are normal parts of aging. However, the problem is not properly addressed in Ethiopia. This study aimed to determine the prevalence and associated factors of depression among the older adults in Bahir Dar city.

### Methods

A community-based cross-sectional survey was conducted among 423 older adults in Bahir Dar city. A simple random sampling technique was used to select the study participants. Depression was assessed using a 15-item Geriatric Depression Scale (GDS). A multivariable logistic regression analysis was used to explore the potential determinants of depression among the participants.

### Results

The prevalence of depression among older adults was found to be 57.9% (95% CI: 53.2–62.6). This study showed that educational status with grades 5-8th (AOR: 5.72, 95% CI: 2.87–11.34), and 9-12th grade (AOR: 3.44, 95% CI: 1.59–7.41), income <2004 ETB (AOR = 1.89, 95% CI: 1.16–3.07), cognitive impairments (AOR: 3.54, 95% CI: 2.16–5.81), family history of mental illness (AOR:3.06, 95% CI: 1.03–9.04), and poor quality of life (AOR: 2.78, 95% CI: 1.74–4.46) were significantly associated with depression.

### Conclusion

The prevalence of depression among older adults was found to be huge. Having low educational status, low monthly income, cognitive impairments, family history of mental illness, and poor quality of life were associated with depression. Therefore, raising community awareness of mental health, increasing social participation, providing supportive counseling

**Data Availability Statement:** All relevant data are within the paper and its Supporting information files.

**Funding:** This study was funded by Bahir-Dar University. The funders had no role in study

design, data collection and analysis, decision to publish, or preparation of the manuscript.

**Competing interests:** The authors have declared that no competing interests exist.

**Abbreviations:** AOR, Adjusted odds ratio; CI, Confidence interval; DSM 5, Diagnostic and Statistical Manual 5ᵗʰ text revision; ETB, Ethiopian birr; GDS, Geriatric Depression Scale; HADS, Hospital anxiety and depression scale; HIV/AIDS, Human immune deficiency syndrome; KM, Kilometer; OR, Odds ratio; SPSS, Statistical Packaging for Social Science; USA, United States of American; WHO, World Health Organization.

and routine screening of depressive symptoms are essential in combating depression among Bahir-Dar city older adults.

## Introduction

According to the Diagnostic and Statistical Manual of Mental Disorders (DSM-5), depression is a common psychiatric condition usually characterized by sadness, lack of interest, guilt or low self-esteem, disturbed sleep or food, exhaustion, and poor attention for at least two weeks [1]. Depression is the most frequent mental health disorder in the world, and it is a serious public health concern because it affects so many people including older adults [2, 3]. It has a prevalence rate of 10 to 55% [4].

Depression in older adults often goes untreated because people typically think that it is a normal component of the aging process and a natural reaction to chronic diseases, loss, and social conversion [2]. Depressive disorders afflict 10 to 20% of older individuals globally, affecting over 300 million people in 2015 as reported by WHO [5]. In addition to that, the aging population is on the rise in many countries of the world. By 2050, it is anticipated that 80% of the world older adults would live in low and middle-income nations, with the number of individuals aged 60 and above reaching 390 million [6]. When compared to their younger counterparts, older persons are more likely to face significant challenges in terms of financial loss, reliance on others, social deprivation, loss of self-worth, and functional limitations. They also have physical and mental health issues [7].

According to numerous studies showed, the prevalence of depression in older adults (aged 60 and above) in Chitradurga was 60% [8], while in Womberma District, Ethiopia, it was 45%. [9] South Africa 40% [10], North Indians 9.5% [11], Vietnam 6.9% [12], Egypt 44.4% [13], Malaysia systematic 16.5% [3], Singapore [14] 13.4%, Ethiopia (Harer) 28.5% [15], Ambo 41.8%, Chinese tertiary hospital 32.8% [16], Tanzania 21.2% [17], Greece 58.5% [18], systematic review conducted in Asian countries 38.6% [19] and Thailand 18.5% [20]. Suicide risk is higher among older adults when they are depressed. Suicide is the most common complication of depression, killing an estimated one million individuals each year [5]. The usage of health services by older adults increases as a result of depression, putting additional strain on the already overburdened healthcare system.

Genetic susceptibility, chronic disease and disability, pain, frustration with limitations in activities of daily living (ADL), personality traits (dependent, anxious, or avoidant), and adverse life events (separation, divorce, bereavement, poverty, social isolation) are all factors that increase depression risk in older adults, according to the WHO [7]. Also, many studies have shown a link between depression and various risk factors such as being a woman, living in a city, insomnia, older adults who are dependent on others, life stressors, lack of a spouse, lack of formal education, lower income, substance abuse, stressful life events, poor social support, more disability, lower life satisfaction, cognitive decline, employment status, and medical comorbidities [9–11, 13, 15, 16, 20–22].

Understanding the epidemiology of depression in older persons is crucial to lessen the harmful impact of depression on daily functioning and quality of life (QOL) [5]. Depression is largely ignored in healthcare strategy and planning in most underdeveloped nations, and mental health services receive only a small amount of funding [19]. In primary care settings, it is both underdiagnosed and undertreated. To the best of the authors' knowledge, the burden of depression among the older adults in Ethiopia has not been adequately investigated, particularly among those living in Bahir Dar city. This gap may contribute to poor or inconsistent

mental health care at the community level. As a result, this research was carried out to estimate the prevalence of depression in the older adults and to investigate the epidemiological factors that contribute to it.

## Method and materials

### Study design, study area and period

A community-based cross-sectional study was conducted in Bahir Dar city administration, which is located 565 km from Addis Ababa in North West of Ethiopia; the capital of Amhara regional state from June 1 to 30, 2020. According to the 2016–17 city administration report, the total population of Bahir Dar city administration is 266, 952; 124,396 males and 142,555 females. The city has nine sub-cities with 66,628 households. Among these, the age group of 60 years and above is estimated to be 11,034 (5003 male, and 6031 females). Those older adults are Shimbit (1670), Tana (1043), Fasilo (1200), Sefene selam (287), Gishabay (522), Shum ambo (417), Belay Zeleke (1591), and Ginbot-20 (4304). The health care service is provided by two specialized hospitals, one specialized and four primary private hospitals. There are also eleven health centers in Bahir Dar city administration. Information is taken from Bahir Dar city municipality.

**Study population, inclusion, and exclusion criteria.** All individuals older adults in the city aged 60 and above and residents of the city for at least six months were included, while older adults people who were severely ill, unable to communicate, and older adults with education below fifth grade were excluded from the study.

### Sample size determination and sampling technique

The sample size was calculated using the single population proportion formula with the assumption of a prevalence (P) of depression of 47.5% from a previous study [23] with a confidence limit of 5%. As a result, n = 384, with no requirement for a correction factor because the population size is more than ten thousand. The ultimate sample size was 423 after adding a 10% non-response rate. The Bahir Dar city administration urban division has nine sub-cities, one of which (Hidar 11) was excluded from the report due to insufficient data. Based on the population size, the final sample size was distributed proportionally to eight sub-cities. Participants included Shimbit (64), Tana (40), Fasilo (46), Sefene Selam (11), Gish Abay (20), Shum Abbo (16), Belay Zeleke (61), and Gimbot Haya (165). The sample frame (households with respective old ages) was obtained from health extension workers, and each household was then randomly selected using the lottery method. If more than one member fulfilled the criteria in one household one was selected using the lottery method. If no participants in the selected household fulfilled the criteria the next household was selected.

### Operational definition

**Older adults.** Are those aged is 60 and above [24].

**Neurocognitive impairment.** The MMSE score of $\leq 22$ for those who attended less than eighth grade, $\leq 24$ for those who attended grade nine to twelve, and $\leq 26$ for those college/university graduates out of a total score of 30 [25].

**Quality of life.** Using WHOQOL-BREF 26-item index; a higher score denotes a higher quality of life [26].

**The multidimensional scale of perceived social support scale.** Any mean scale score ranging from 1 to 2.9 is considered as low, 3 to 5 moderate social support, and 5.1 to 7 as high social support [27].

**Activities.**  Using Katz's indicator of daily life independence, a total score of six shows independence, four suggests moderate independence, and two or less implies dependence [28].

**Nutritional status.**  was assessed using the Mini Nutritional Assessment with the score ranged from 0–14 interpreted as (0–7) as malnourished, (8–11) at risk for malnutrition, and (12–14) as normal nutritional status [29].

**Current substance use.**  Within the last three months, you have used at least one of a certain substance for non-medical purposes (alcohol, khat, tobacco, others) [30].

**Ever use of a substance.**  Using at least one of any specific substances for a non-medical purpose at least once in a lifetime (alcohol, khat, tobacco, others [30].

**Fast Alcohol Screening Test (FAST).**  An overall total score of 3 indicates hazardous alcohol consumption [31].

**Income.**  Using the World Bank poverty line cut point those who have an average monthly income of less than 2004 ETB (1.9 USD/day) taking 1$ = 35.16 ETB were taken as low income [32].

## Data collection tool

A structured interviewer-administered questionnaire was used to assess the sociodemographic factors, clinical related factors, behavioral and psychosocial factors. Geriatric Depression Scale (GDS-15) item was used to determine whether elderly people had depression or not. GDS-15 has undergone rigorous testing and validation in low- and middle-income countries including India and Nepal [33, 34].

The Royal College of Physicians, the British Geriatric Society, and the Royal College of General Practitioners all recommended this geriatric depression scale for screening depression in older adults [35]. A cutoff value of more than or equal to five was used to define depression [36]. The internal consistency (Cronbach alpha) of GDS-15 in this study was 0.86. Cognition status using standardized mini-mental state examination (MMSE) with a cut-off point as follows, no cognitive impairment (24–30), mild cognitive impairment (18–23), severe cognitive impairment (0–17) (39). It has been validated in Ethiopia for those with a formal education grade of fifth or higher, with different cut-off points depending on their level of education [25]. The specificity and sensitivity of MMSE were 77.8% and 78.7% respectively [37]. WHO-QOL-BREF was used to assess the quality of life. It has four domains; physical, psychological, social, and environmental factors. It has internal consistency (Cronbach's alpha $> = 0.7$) and has been translated into nine languages [26]. The six-question mini-nutritional assessment short form has been validated in Ethiopia. Cronbach's alpha was 0.65, with 80.1 percent sensitivity and 72.5 percent specificity (54). Katz's index of daily life independence consists of six questions, each worth one point. Cronbach's alpha was found to be 0.83, with strong test-retest and inter-rater reliability [38]. History of mental illness presence of chronic medical illness, and substance-related factors were assessed with yes/no questions, but alcohol drinking was assessed by using FAST.

## Data collection procedure

Data was collected through face-to-face interviews by trained data collectors. The data were collected from study participants by face-to-face interviews from house to house. The questionnaire was prepared in English and then translated into the local language (which is Amharic) by a language translator and translated back to English to ensure its understandability and consistency before the actual data collection. The training was given for the supervisor and data collectors by the principal investigator for two days duration on the methods of data collection

and the detail of the questionnaire. Data were collected by four psychiatric nurses who currently work in health centers and was supervised by Masters of Sciences degree holder in mental health. A pretest was conducted on 21(5%) to check the understandability of the questionnaires. The collected data were reviewed and checked for completeness before data entry.

### Data analysis

The completed questionnaire was manually checked for completeness. Data were coded and entered into Epi data version 4.6 and, then exported to SPSS- 26 version for analysis. Descriptive and summary statistics were used to explain the population concerning the relevant variables. The bivariate logistic analysis was done to determine the association between the outcome and explanatory variables. Variables with p less than 0.25 in the bivariate analysis were entered into multivariate analysis. Multivariable logistic regression analysis was employed to control for possible confounding effects and to determine the presence of a statistically significant association between independent variables and outcome variables. The model of fitness was checked by Hosmer and Lemeshow goodness and a p-value less than 0.05 was considered statistically significant and the strength of the association was presented by an odds ratio of 95% C.I.

### Ethical consideration

Ethical clearance was obtained from the Institutional Review Board of Bahir Dar University. Study participants were informed about the procedure, the significance of the study, risks, and benefits associated with the study. Written Informed consent was obtained from participants who participated in the study. Each respondent was informed about the objective of the study and all data obtained from them was kept confidential by using code instead of any personal identifier which was used only for the study. The information was not disseminated without the respondent's permission. The information provided by the participants was exclusively utilized for the study. Those older adults who reported depression were immediately referred to mental health facilities for further evaluation and management.

## Results

### Socio-demographic characteristics of participants

A total of 423 older adult individuals participated in this study (100% of response rate). The mean age (SD) of the participants was 66.01(±5.88), 58.9% were males. The majority of the participants 52.2% had a spouse. Almost two-thirds of the participants 66.6% were Orthodox Christian followers and the majority 86.3% were Amhara in their ethnicity. Regarding educational level, 61.5% were from grade five to eight. Around one-third, 36.6% were housewives followed by retired 27.0%. From the participants, 62.9% participants reported that their average monthly income was ≥2004 ETB, and 63.6% were living with their family (Table 1).

### Clinical and substance-related factors of the participants

According to this study finding, 42.1% of respondents had neurocognitive impairment. More than half, 51.1% of participants had a comorbid medical illness, such as hypertension 30.7%, HIV/AIDS 6.1%, cardiac 5.9%, diabetes 18%, and others 2.6%. Of the participants, 46.8% currently used medication and 8.0% had reported a family history of mental illness. Among the respondent, 11.8% of the respondents had a history of head trauma and 53.4% were normal nutritional status (Table 2).

**Table 1. Socio-demographic characteristics of older adults in Bahir Dar city administration, northwest, Ethiopia, 2020 (n = 423).**

| Variables | Categories | Frequency(n = 423) | Percent (%) |
|---|---|---|---|
| Age in years | 60–64 | 191 | 45.1 |
| | 65–69 | 127 | 30.0 |
| | 70–74 | 69 | 16.3 |
| | 75–79 | 18 | 4.3 |
| | 80 and above | 18 | 4.3 |
| Sex | Female | 174 | 41.1 |
| | Male | 249 | 58.9 |
| Marital status | Has spouse | 221 | 52.2 |
| | No spouse | 202 | 47.8 |
| Religion | Orthodox | 282 | 66.6 |
| | Muslim | 117 | 27.7 |
| | Protestant | 16 | 3.80 |
| | Catholic | 8 | 1.90 |
| Ethnicity | Amhara | 365 | 86.3 |
| | Oromo | 14 | 3.3 |
| | Tigre | 27 | 6.4 |
| | Gurage | 17 | 4 |
| Educational status | 5-8th grade | 260 | 61.5 |
| | 9-12th grade | 99 | 23.4 |
| | College and above | 64 | 15.1 |
| Occupational status | Governmental employee | 17 | 4.0 |
| | Merchant | 105 | 24.8 |
| | Housewife | 155 | 36.6 |
| | Retired | 114 | 27.0 |
| | Others* | 32 | 7.6 |
| Monthly income | <2004ETB | 157 | 37.1 |
| | ≥2004 ETB | 266 | 62.9 |
| Current living condition | Alone | 92 | 21.7 |
| | Relative | 62 | 14.7 |
| | Family | 269 | 63.6 |

Key: * Farmer, Jobless.

Regarding the use of the substance, about 9.7% of them were hazardous alcohol users. Of the participants, 8% and 2.4% were using khat and cigarettes within the past three months respectively (Fig 1).

## Psychosocial characteristics of respondents

Regarding the psychosocial factors of respondents around 47.5% of them had poor quality of life. From the participant, 13.0%, 13.2%, and 13.5% had low family social support, low friend social support, and low other social support respectively (Table 3).

## Prevalence of depression and associated factors among older adults people

In this study, the overall prevalence of depression among older adults people in Bahir Dar city was 57.9% (95% CI: 53.2,62.6) (Fig 2).

Factors such as marital status, educational status, occupational status, monthly income, current living condition, cognitive impairment, history of chronic medical illness, medication

**Table 2. Clinical characteristics of older adults people in Bahir Dar city administration, northwest, Ethiopia, 2020 (n = 423).**

| Variables | Categories | Frequency(n = 423) | Percent (%) |
|---|---|---|---|
| Cognitive impairments | Yes | 178 | 42.1 |
| | No | 245 | 57.9 |
| History of chronic medical illness | Yes | 216 | 51.1 |
| | No | 207 | 48.9 |
| Hypertension | Yes | 130 | 30.7 |
| | No | 87 | 20.6 |
| HIV/AIDS | Yes | 26 | 6.1 |
| | No | 192 | 45.4 |
| Cardiac | Yes | 25 | 5.9 |
| | No | 192 | 45.4 |
| Diabetes | Yes | 76 | 18.0 |
| | No | 141 | 33.3 |
| Others* | Yes | 11 | 2.6 |
| | No | 207 | 48.9 |
| Medication currently in use | Yes | 198 | 46.8 |
| | No | 225 | 53.2 |
| Family history of mental illness | Yes | 34 | 8.0 |
| | No | 389 | 92.0 |
| History of head trauma | Yes | 50 | 11.8 |
| | No | 373 | 88.2 |
| Nutritional status | Malnourished | 32 | 7.6 |
| | Risk | 165 | 39 |
| | Normal | 226 | 53.4 |

Others*: Epilepsy.

currently in use, family history of mental illness, and quality of life were significantly correlated ($P < 0.25$) in bi-variable analysis. Among these, variables such as educational status, income, cognitive impairment, family history of mental illness, and poor quality of life were significantly associated with depression in multivariable analysis.

Older adults whose educational status was grades 5-8th were nearly six times (AOR: 5.72, 95% CI: 2.87–11.34), and 9-12th grade were 3.44 times (AOR: 3.44, 95% CI: 1.59–7.41) more likely to develop depression compared to college and above. Older adults with a monthly income of <2004 ETB were nearly 2 times more likely to have depression as compared to participants with an income of <2004 ETB (AOR = 1.89, 95% CI: 1.16–3.07). Older adults who had cognitive impairments were 3.54 times more likely to develop depression compared with their counterparts (AOR: 3.54, 95% CI: 2.16–5.81), and, who had a family history of mental illness were also three times more likely to have depression compared to those who had no family history of mental illness (AOR:3.06, 95% CI: 1.03–9.04). Finally, older adults with poor quality of life were 2.78 times more likely to develop depression compared to good quality of life (AOR: 2.78, 95% CI: 1.74–4.46) (Table 4).

## Discussion

The high prevalence of depression in this study 57.9% [95% CI: 53.2% − 62.6%], may be the indicative of a high burden due to depression among older adults in the community. The finding was congruent with that of a community-based cross-sectional study done in Chitradurga,

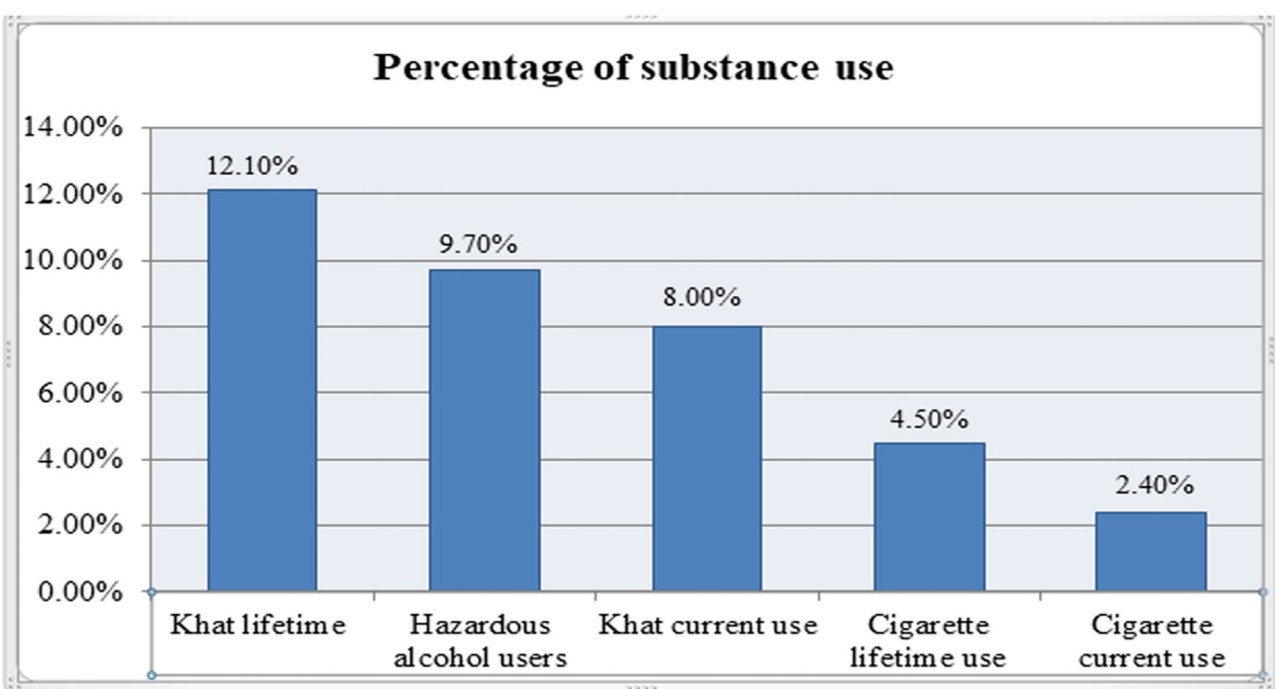

**Fig 1. Ever and current substance use among older adults people in Bahir Dar city administration, northwest, Ethiopia, 2020 (n = 423).**

India (60%) [8], Heraklion, Greece (58.5%) [18], Portugal (61.4%) [39], and India (53.75%) [40]. This result was lower than those found in studies in Greece (84.3%) [41], Vietnam (66.9%) [12], urban, India (75.5%) [42], and Beni Suef, Egypt (89.7%) [43]. This disparity in prevalence could be related to differences in the tools employed to measure depression. For instance, in urban India, the 30-item GDS is used to screen for depression, whereas in Vietnam, the Zung self-rating depression scale is used to screen for depression [12]. Furthermore, the heterogeneity in the prevalence of depression among older adults could be explained by differences in study design, sampling procedure, socioeconomic-demographic characteristics, geographical location, and cultural differences.

**Table 3. Psychosocial characteristics of older adults people in Bahir Dar city administration, northwest, Ethiopia, 2020 (n = 423).**

| Variables | Categories | Frequency(n = 423) | Percent (%) |
|---|---|---|---|
| The activity of daily living | Dependent | 9 | 2.1 |
| | Moderate | 14 | 3.3 |
| | Independent | 400 | 94.6 |
| Quality of life | Poor quality | 201 | 47.5 |
| | Good quality | 222 | 52.5 |
| Family social support | Low support | 55 | 13.0 |
| | Moderate support | 136 | 32.2 |
| | High support | 232 | 54.8 |
| Friend social support | Low support | 56 | 13.2 |
| | Moderate support | 213 | 50.4 |
| | High support | 154 | 36.4 |
| Significant other social support | Low support | 57 | 13.5 |
| | Moderate support | 150 | 35.5 |
| | High support | 216 | 51.1 |

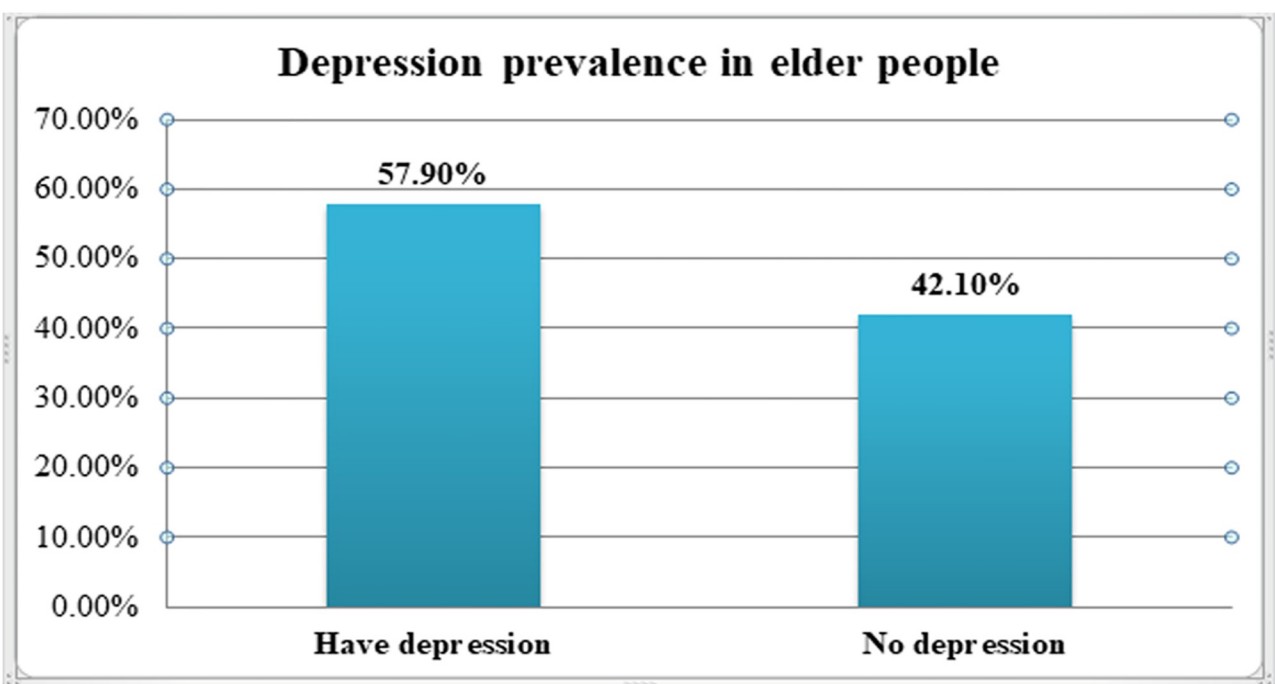

**Fig 2. Prevalence of depression in older adults people in Bahir Dar city administration, northwest, Ethiopia, 2020 (n = 423).**

However, this study finding was higher than study done in North Indian (9.5%) [11], Malaysian (16.5%) [3], Tanzania 21.2% [17], a systematic review conducted in China (38.6%) [19], Thailand (18.5%) [20], Ambo, Ethiopia (41.8%) [21], Singapore 13.4% [14], China 32.8% [16], and Womberma district, Ethiopia (45%) [9]. This variation might be due to social-cultural, economic disparities, and the heterogeneity in the classification of depression, i.e., they utilized a GDS-15 score of 6 and above to define depression, which could lead to an underestimating of depression prevalence in Chinan [19]. Another probable reason is the difference in assessment technique; in Singapore, depression was assessed using the Geriatric Mental State (GMS) instrument [14]. Whereas, in our study, depression was assessed using the Geriatric Depression Scale item 15 (GDS-15) tool. Additionally, the disparity could be due to a difference in the study participants; in Ambo, the majority of the participants were males, which was found to be less likely to be depressed than females in the study [21]. Furthermore, according to some studies, people in developed countries have easier access to mental health care and support before they experience problems.

Regarding the associated factors, older adults whose educational status grades 5-8[th] were nearly 6 times and 9-12th grades were 3.44 times more likely to develop depression compared to college and above. This finding was in agreement with different studies in Ethiopia (Harer) [15], Malaysia [3], India (Punjab) [22], Egypt [44], and Thailand [20]. Depressive symptoms are linked to educational attainment, and depression can be influenced by a variety of socio-economic factors. In lower levels of educational achievement, there is no simple strategy to improve the health and economic success of a nation.

Older adults with a monthly income <2004 ETB were nearly 2 times more likely to have depression as compared to participants with an income of ≥2004 ETB. Similar to a finding of different studies reported in Asia (Myanma) [45], North Indians [11], and Portugal [46]. This is the finding that low-income people have more difficult getting healthy services and care,

**Table 4. Bivariate and multivariable logistic regression analysis results of depression in Bahir Dar city administration, northwest, Ethiopia, 2020 (n = 423).**

| Variables | Category | Depression | | COR(95%C.I) | AOR(95%C.I) | P-values |
|---|---|---|---|---|---|---|
| | | Yes(n) | No(n) | | | |
| Marital status | No spouse | 145(71.8%) | 57(28.2%) | 3.08(2.05,4.62) | 1.49(0.92,2.44) | 0.108 |
| | Has spouse | 100(45.2%) | 121(54.8%) | 1 | 1 | |
| Educational status | 5-8th grade | 176(67.7%) | 84(32.3%) | 5.79(3.14,10.69) | 5.72(2.87,11.34) | <**0.001**$^*$ |
| | 9-12th grade | 52(52.5%) | 47(47.5%) | 3.06(1.55,6.04) | 3.44(1.59,7.41) | **0.002**$^*$ |
| | College and above | 17(26.6%) | 47(73.4%) | 1 | 1 | |
| Occupational status | Government employee | 6(35.3%) | 11(64.7%) | 1 | 1 | |
| | Merchant | 52(49.5%) | 53(50.5%) | 1.79 (0.62, 5.22) | 0.96(0.25,3.66) | 0.949 |
| | Housewife | 100(64.5%) | 55(35.5%) | 3.333(1.170,9.50) | 1.33(0.35,5.04) | 0.669 |
| | Retired | 61(53.5%) | 53(46.5%) | 2.11(0.73, 6.09) | 1.41(0.37,5.28) | 0.612 |
| | Others** | 26(81.3%) | 6(18.8%) | 7.94(2.09, 30.13) | 1.71(0.34,8.63) | 0.516 |
| Monthly income | <2004ETB | 112(71.3%) | 45(28.7%) | 2.49(1.63,3.79) | 1.89(1.16,3.07) | **0.01**$^*$ |
| | ≥2004 ETB | 133(50.0%) | 133(50.0%) | 1 | 1 | |
| Current living condition | Alone | 65(70.7%) | 27(29.3%) | 2.22(1.33, 3.69) | 0.83(0.39,1.75) | 0.634 |
| | Relative | 40(64.5%) | 22(35.5%) | 1.67(0.94, 2.97) | 0.46(0.21,1.03) | 0.061 |
| | Family | 140(52.0%) | 129(48.0%) | 1 | 1 | |
| Cognitive impairment | Yes | 140(78.7%) | 38(21.3%) | 4.91(3.16,7.62) | 3.54(2.16,5.81) | <**0.001**$^*$ |
| | No | 105(42.9%) | 140(57.1%) | 1 | 1 | |
| History of chronic medical illness | Yes | 139(64.4%) | 77(35.6%) | 1.72(1.16,2.54) | 1.17(0.74,1.86) | 0.492 |
| | No | 106(51.2%) | 101(48.8%) | 1 | 1 | |
| Medication currently in use | Yes | 130(65.7%) | 68(34.3%) | 1.83(1.23,2.71) | 1.04(0.36,3.02) | 0.943 |
| | No | 115(51.1%) | 115(51.1%) | 1 | 1 | |
| Family history of mental illness | Yes | 29(85.3%) | 5(14.7%) | 4.64(1.76,12.25) | 3.06(1.03,9.04) | **0.043**$^*$ |
| | No | 216(55.5%) | 173(44.5%) | 1 | 1 | |
| Quality of life | Poor quality | 153(76.1%) | 48(23.9%) | 4.50(2.96,6.85) | 2.78(1.74,4.46) | <**0.001**$^*$ |
| | Good quality | 92(41.4%) | 130(58.6%) | 1 | 1 | |

$^*$Statistically significant at P-value < 0.05, COR, Crude odds Ratio, AOR, Adjusted odds Ratio, 1 = reference category, Chi square = 8, Hosmer Lemeshow goodness-of-fit 0.42, degrees of freedom = 8 and,

** Farmer, Jobless.

which has been associated with higher levels of depression. McCall and colleagues' findings in the United States supported prior studies that connected low income to a higher prevalence of depression [47].

Older adults who had cognitive impairments were 3.54 times more likely to develop depression compared with their counterparts. This was supported by the study conducted in Ethiopia (Harer) [15] and Chinese tertiary hospitals [16]. According to Ismail's meta-analysis, depression is common among people with mild cognitive impairment (MCI), with a pooled prevalence of 32% [48]. Depression can result from problems with attention and working memory, as well as changes in sleep patterns and social isolation due to cognitive impairment [49]. Furthermore, MCI shares some of the same characteristics as late-life depression in terms of brain structure changes [50].

A family history of mental illness was also a predictor of depression. When compared to respondents who did not have a family history of mental illness, those populations who had a family history of mental illness were three times more likely to be depressed. This could be explained by the fact that mental illness is inherited, that families are stigmatized, and that there are various types of burdens on family members in terms of financial expenses and

providing care for the patient, as well as the offspring may be stressed and worried about their parent's health condition, all of which could increase the risk of depression [51].

Finally, older adults with poor quality of life were 2.78 times more likely to develop depression compared to good quality of life. This matches research from North Indians [11], Chinese tertiary hospitals [16], and Portugal [46]. Our findings show that older people with depression are more likely to report poor quality of life. In a review article comprising 74 studies, Sivertsen and colleagues came to the same conclusion, finding that depressed older adults had a lower global quality of life than non-depressed older adults. They went on to say that this link remained constant throughout time and was irrespective of how the quality of life was measured [52].

## Limitations

The limitation of our study is the use of the GDS scale to measure depressive symptoms rather than formal interviews for diagnosing depression, which is thought to be more appropriate for identification and less sensitive to somatic symptoms that could lead to the overestimation of depression. Other limitations include, some of the reports were based on prior events, which can lead to recall bias. Variables like alcohol use, khat chewing, and other substances are more sensitive issues that might lead to social desirability bias. Also, the generalizability of the study might be limited for those who had formal education since the tool (MMSE) is adapted based on the educational level in this setup. In addition, Since income was assessed using the World Bank poverty line, it has limitations such as simple and does not take into account indebtedness, health, education, housing, or public service access. And it does not always fully reflect the differences in subsistence costs between countries. Finally, because of the nature of the cross-sectional study design, it is impossible to establish cause and effect linkages.

## Conclusion

The study conducted in Bahir Dar city shows that more than half of the older adults are suffering from depressive symptoms. An older adult having low educational status, low income, cognitive impairments, a family history of mental illness, and a poor quality of life were all found to be significant predictors of depression in older adults. Because geriatric depression is sometimes unrecognized by clinicians and depressive symptoms are often attributed to the aging process, we recommend that clinicians regularly screen depressive symptoms using standard assessment tools in health care settings and the community. It is preferable to place a greater emphasis on the risk groups identified by this finding.

## Supporting information

**S1 File.**
(XLS)

## Acknowledgments

We would like to thank the Bahir Dar university department of psychiatry. Also, we want to thank participants and data collectors for their willingness to be part of the study.

## Author Contributions

**Conceptualization:** Tamrat Anbesaw.

**Data curation:** Tamrat Anbesaw, Betelhem Fekadu.

**Formal analysis:** Tamrat Anbesaw.

**Funding acquisition:** Betelhem Fekadu.

**Investigation:** Tamrat Anbesaw, Betelhem Fekadu.

**Methodology:** Tamrat Anbesaw, Betelhem Fekadu.

**Resources:** Betelhem Fekadu.

**Software:** Tamrat Anbesaw.

**Supervision:** Betelhem Fekadu.

**Visualization:** Tamrat Anbesaw, Betelhem Fekadu.

**Writing – original draft:** Tamrat Anbesaw.

**Writing – review & editing:** Tamrat Anbesaw.

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
