## [Decision Letter · Decision Letter 0]

20 Jan 2022

PONE-D-21-38673Prevalence of depression and associated factors among elderly in Bahir Dar city administration, Northwest Ethiopia, 2020: Cross-sectional studyPLOS ONE

Dear Tamrat Anbesaw,

Thank you for submitting your manuscript to PLOS ONE. After careful consideration, we feel that it has merit but does not fully meet PLOS ONE’s publication criteria as it currently stands. Therefore, we invite you to submit a revised version of the manuscript that addresses the points raised during the review process.

This study has the potential to provide the invaluable prevalence of depression and its associated factors among older adults in Ethiopia, but authors should carefully address all comments provided by reviewers as per the below instruction for further review.  In addition to the reviewer's suggestions please remove abstract lines 35-40 big bracket [ ] and only keep small bracket and follow in discussion line 301 too. Submit manuscript with full proofread. 

We look forward to receiving your revised manuscript.

Kind regards,

Sharada Prasad Wasti, Ph.D., MSc, MHCM, MA.

Academic Editor

PLOS ONE

Journal Requirements:

 [The funders had no role in study design, data collection and analysis, decision to publish, or preparation of the manuscript.]

[NO authors have competing interests.]

Address all below comments and suggestions:

Reviewer #1: The authors determined the prevalence and associated factors of depression among 423 older adults in Bahir Dar city. Depression was assessed by 15-point Geriatric Depression Scale, where older adults scoring five or more where classified as being depressed. Cognitive impairment, quality of life and income were assessed by MMSE, WHOQOL-BREF 26-item index and World Bank poverty line cutoff respectively. The prevalence of depression was found to 57.9 %. The authors found that depression was associated with cognitive impairment, poor quality of life and low income.

This study has a potential to provide invaluable epidemiological data for depression among the older adults in Ethiopia, provided the following issues are addressed.

Main Issues

1. Prevalence calculation.

The authors did not explain how the prevalence of depression among older adults was calculated. The data for the total population of older adults aged 60 years and above in Bahir Dar City is not provided. The total number of households in Bahir Dar city is not provided either. It is not clear how the 423 subjects were obtained. The 423 older adults who participated in this study may not be representative of the older adults in Bahir Dar City. The study findings (particularly the prevalence) cannot therefore be generalized to the older adults in Bahir Dar City. This is hardly addressed in the discussion.

2. Validation of Assessment Tools

The authors did not state whether GDS has been validated in Ethiopia and whether they used validated versions of GDS. The limitations of assessing income using World Bank Poverty line cut-offs are not appreciated. The limitations of MMSE in assessing cognitive impairment are not addressed. The authors did not state whether older adults who were illiterate or had level of education below fifth grade were excluded from the study.

Other issues

Title.

The title needs to be reframed as the study population is not representative of the older adults in Bahir Dar City.

Abstract

Line 27 Depression is unnoticed by clinicians due to natural aging process. This needs to be reviewed. There are several factors for under-diagnosis of depression in older adults, apart from issues related to aging.

Line 29 …elderly in Bahir Dar city residents?

Line 35 use older adults instead of elders

Lines 43, 44 consider restating the concluding statement to reflect your findings. Emotional care, mental health care, what is the difference?

INTRODUCTION

Lines 52, 53. The statements are incompatible with what the authors are studying.

Line 56 re-check whether WHO information on depression includes two weeks

Lines 56, 57, 58 repetition of information

Lines 61,62 consider reframing the statement.

Line 65 aged 60 and up??

Lines 69 to 77. Consider rearranging this paragraph. The magnitude of depression in older adults should appear before the consequences of depression. The authors should be specific regarding the age group the prevalence estimates of depression refer to.

Line 83 consider reframing this statement…..elderly people who are reliant on others vs dependence on others, reliance on others??

Methods and materials

Line 113, 114 What is the total population of older adults aged 60 years and over in Bahir Dar City? Unless this is known it will be impossible to estimate the prevalence of depression among older adults in Bahir Dar City that can be generalized to the whole population of older adults in Bahir Dar City.

Line 119 Which information was taken from Bahir Dar City Municipality?

Line 124 All old age residents?? Consider correcting this statement.

Line 126 All old age population?? Same as in Line 124

Line 128 Consider correcting this sentence.

Line 133, 134 consider reframing the statements. Was the sample size estimated using Epi info 7 Stat Calc software or single population proportion formula?

Line 141 Which population size are you referring to here; the total population or the population of older adults aged 60 and over?

Line 142 Clarify how the final sample size was distributed proportionally among the eight sub-cities.

Line 144 sample frame? od ages? Correct.

Line 151 Limitations of MMSE should be addressed

Line 156 Has this instrument been validated in Ethiopia?

Line 170 The limitations of assessing income using this method should be addressed.

Line 176 Has GDS been validated in Ethiopia ? Which version of GDS was used? Validated or original version?

Line 182 What about the older adults with education below fifth grade, and those who could neither read nor write? Were they excluded from the study? The limitations of using MMSE to assess cognition should be appreciated. How was dementia distinguished from depression in this study?

Lines 200,201.202 , 204 Consider correcting these statements. What is MSc?

Lines 202, 203,204 Was the GDS translated into the local language(Hamharic)?

Lines s 207, 2008,209. Consider correcting these statements.

Line 222 How was the confidence interval of the prevalence calculated?

Line 235 How about the older adults with level of education below the fifth grade and those who could neither read nor write? Were they excluded from the study?

Line 265 correct this statement.

Line 271 Show how the prevalence of depression was calculated. How was the confidence interval calculated?

Lines 283 to 292 Consider correcting the statements with 3.44 times, 2.78 times etc.

Line 313 to 328 Consider summarizing this paragraph.

Line 334 Low education may limit the development of therapies to alleviate the disease burden of depression. Can older adults develop therapies for depression? Please consider correcting this statement.

Lines 339 to 342 Consider correcting the statements.

Line 343 correct 3.54 times

Line 349 limitations of MMSE in assessing cognitive impairment should be addressed.

Line 352 populations who had family history of mental illness? Consider correcting this.

Line s 376 to 382 Correct the statements.eg we recommend

that clinicians regularly screening depressive symptoms using standard studies in the elderly

Line 399 What about the older adults who were illiterate? Were they excluded from the study? Were there older adults who were unable to provide informed consent due to cognitive impairment? How many older adults had severe cognitive impairment per MMSE?

Line 404 Any time during the procedure?

Lines 404 ,405 correct the statements.

Typos and Grammatical errors

There several typos and grammatical errors in this manuscript apart from those highlighted . These should to be corrected

Reviewer #2: The authors have presented findings which adds to current literature on depression. Overall the methods chosen were appropriate and the results support the conclusions. However, the authors need to improve on the language of the manuscript.

---

## [Author Response · Author response to Decision Letter 0]

2 Feb 2022

Response to reviewer’s 

Sharada Prasad Wasti, Ph.D., MSc, MHCM, MA, Academic Editor, PLOS ONE.

Thank you very much for giving us the golden chance to revise our manuscript “Depression and associated factors among older adults in Bahir Dar city administration, Northwest Ethiopia, 2020: Cross-sectional study.

We would also like to thank the reviewers for their detailed reviews and for providing us with helpful suggestions that will strengthen our manuscript and knowledge. We have gone through the comments and tried to include the responses to all the comments and suggestions. Also, we addressed questions raised by the academic editor. 

Comments by editor and reviewers

 This study has the potential to provide the invaluable prevalence of depression and its associated factors among older adults in Ethiopia, but authors should carefully address all comments provided by reviewers as per the below instruction for further review. 

Response: We tried to revise and incorporate carefully based on the editor and reviewers' feedback.

 In addition to the reviewer's suggestions please remove abstract lines 35-40 big bracket [ ] and only keep small bracket and follow in discussion line 301 too. Submit manuscript with full proofread. 

Response: We removed it from the manuscript as recommended.

Response: We prepared it as recommended.

 If you would like to make changes to your financial disclosure, please include your updated statement in your cover letter

Response: We amended it in the revised cover letter.

Journal Requirements

Response: We try to put the manuscript to meet PLOS ONE's style requirements.

 [The funders had no role in study design, data collection and analysis, decision to publish, or preparation of the manuscript.]

Response: This study was funded by Bahir-Dar University. The funders had no role in study design, data collection and analysis, decision to publish, or preparation of the manuscript.

Response: We received only financial support and we received only financial support from our institution Bahir-Dar University.

Response: We rewrite it as “The funders had no role in study design, data collection and analysis, decision to publish, or preparation of the manuscript.”

Response: Corresponding Author

Response: We amended it as per comment.

Response: We revised and sent the cover letter.

[NO authors have competing interests.]

Please complete your Competing Interests on the online submission form to state any Competing Interests. If you have no competing interests, please state "The authors have declared that no competing interests exist."

Response: Corrected as recommended.

Response: We revised and sent the cover letter.

Response: Comment accepted and amended.

Comments by reviewers 

Reviewer 1

The authors determined the prevalence and associated factors of depression among 423 older adults in Bahir Dar city. Depression was assessed by 15-point Geriatric Depression Scale, where older adults scoring five or more where classified as being depressed. Cognitive impairment, quality of life and income were assessed by MMSE, WHOQOL-BREF 26-item index and World Bank poverty line cutoff respectively. The prevalence of depression was found to 57.9 %. The authors found that depression was associated with cognitive impairment, poor quality of life and low income.

This study has a potential to provide invaluable epidemiological data for depression among the older adults in Ethiopia, provided the following issues are addressed.

Main Issues

1. Prevalence calculation.

The authors did not explain how the prevalence of depression among older adults was calculated. The data for the total population of older adults aged 60 years and above in Bahir Dar City is not provided. The total number of households in Bahir Dar city is not provided either. It is not clear how the 423 subjects were obtained. The 423 older adults who participated in this study may not be representative of the older adults in Bahir Dar City. The study findings (particularly the prevalence) cannot therefore be generalized to the older adults in Bahir Dar City. This is hardly addressed in the discussion.

Response: We explained in detail in the next questions in the method sections. 

-Calculating prevalence: The prevalence of depression among older adults is calculated by measuring the presence of depression in a sample of the population selected randomly, then dividing the number of older adults that have depression by the number of people in whom it was measured. Prevalence is often expressed as a percentage.

Simply,

 Prevalence= (Total cases)/(Total population) x 100, P=245/423 x 100 = 57.9%

-The technique of how we addressed the participants is that “Among these, the age group of 60 years and above is estimated to be 11,034 (5003 male, and 6031 females). Those older adults are estimated Shimbit (1670), Tana (1043), Fasilo (1200), Sefene selam (287), Gishabay (522), Shum ambo (417), Belay Zeleke (1591), and Ginbot-20 (4304)”. From the above total older adults we allocated proportionally and from this, Shimbit (64), Tana (40), Fasilo (46), Sefene Selam (11), Gish Abay (20), Shum Abbo (16), Belay Zeleke (61), and Gimbot Haya (165) were included in this study.” 

-We tried to represent the older adult population in Bahir-Dar city administration using probability sampling techniques. We discussed clearly in sampling technique. 

2. Validation of Assessment Tools

The authors did not state whether GDS has been validated in Ethiopia and whether they used validated versions of GDS. The limitations of assessing income using World Bank Poverty line cut-offs are not appreciated. The limitations of MMSE in assessing cognitive impairment are not addressed. The authors did not state whether older adults who were illiterate or had level of education below fifth grade were excluded from the study.

Response: Thank you for your very critical and constructive comments! Geriatric Depression Scale item 15 (GDS-15) was used to assess the presence of depression among older adults. Even if the tool hasn’t been validated in Ethiopia we use extensively validated in low and middle-income countries such as India, Nepal, other Asian and African countries with a sensitivity of 92% and specificity of 89%. Also, various studies conducted in Ethiopia use this instrument to assess depression. The validity and reliability of the tool have been supported through both clinical practice and research. In addition, we conducted a pretest and its internal consistency (Cronbach alpha) in this study was 0.86. Finally, we do have a plan to conduct a validation study for the future. 

-Regarding the World Bank poverty line cut-offs; It is simplistic and does not reflect indebtedness, health, education, housing, or access to public services. Also, it does not always accurately represent the different costs of subsistence from country to country. Difficulty to assess each item found in the household, despite this limitation many developing country studies use this assessment way. It's better to assess the wealth index using PCA for future researchers. W put it in the limitation section. Thank you for your very supportive comment. we incorporate in the limitation section.

-Regarding MMSE, One important limitation of MMSE is that it cannot be administered to illiterate subjects as 2 of its items involve reading and writing. Also, the limitation is the inclusion of a task requiring paper and pencil (copying a drawing). Furthermore, difficulty in identifying mild cognitive impairment. We excluded older adults with education below fifth grade and we revised in the exclusion criteria accordingly. 

Other issues

Title.

The title needs to be reframed as the study population is not representative of the older adults in Bahir Dar City.

Response: It represents the older adults population in B.dar city. We put it clearly in the method section how we did allocate it.

Abstract

Line 27 Depression is unnoticed by clinicians due to natural aging process. This needs to be reviewed. There are several factors for under-diagnosis of depression in older adults, apart from issues related to aging.

Response: We paraphrased it accordingly. Several studies showed, depression among the elderly were unnoticed by the individuals themselves and were also underdiagnosed by healthcare professionals due to the misconception that these are a natural aspect of the aging process and a natural reaction to chronic diseases, loss, and social conversion. It could negatively aggravate several aspects at the individual, household, national, and international level. 

NB: In the clinical setting, older adults with depression have similar manifestations to dementia (we call it pseudodementia), this is considered as an age-related problem and it is ignored by the patient itself, even parents, and underdiagnosed by clinicians. Hence, we rephrase as “Depression is the most common psychiatric condition among older adults, and it goes unnoticed by individuals themselves and is under-diagnosed by clinicians due to the misconception that these are normal parts of aging.”

Line 29 …elderly in Bahir Dar city residents?

Response: We corrected it accordingly. The study represents depression among the elder people Bahir-Dar city. We corrected as” older adults in Bahir Dar city”

Line 35 use older adults instead of elders

Response: Thank you, we corrected it as recommended throughout the manuscript. 

Lines 43, 44 consider restating the concluding statement to reflect your findings. Emotional care, mental health care, what is the difference?

Response: Thank you, we revised and added some additional ideas and we rewrite as “The prevalence of depression among older adults was found to be huge. Educational status, monthly income, cognitive impairments, family history of mental illness, and poor quality of life were associated with depression. To fulfill the demands of the city's growing older population, elderly care, mental health care, and social security services should be strengthened. Also, raising community awareness of mental health, increasing social participation, and providing supportive counseling are essential in combating depression among Bahir-Dar city older adults”. See the main document. 

INTRODUCTION

Lines 52, 53. The statements are incompatible with what the authors are studying.

Response: We removed it. 

Line 56 re-check whether WHO information on depression includes two weeks

Response: Thank you! we changed it according to the Diagnostic and Statistical Manual of Mental Disorders (DSM-5).

Lines 56, 57, 58 repetition of information

Response: We revised it!

Lines 61,62 consider reframing the statement.

Response: Revised, We paraphrased as “Depression in older adults often goes untreated because people typically think that it is a normal component of the aging process and a natural reaction to chronic diseases, loss, and social conversion”. See in the main document.

Line 65 aged 60 and up??

Response: We change to “aged 60 and above.” 

Lines 69 to 77. Consider rearranging this paragraph. The magnitude of depression in older adults should appear before the consequences of depression. The authors should be specific regarding the age group the prevalence estimates of depression refer to.?? See ambo study pop/n

Response: Thank you! We rearranged it as you recommended. And, we specify age group, older adults: are those aged is 60 and above.

Line 83 consider reframing this statement…..elderly people who are reliant on others vs dependence on others, reliance on others??

Response: We corrected as by saying “older adults who are dependent on others”

Methods and materials

Line 113, 114 What is the total population of older adults aged 60 years and over in Bahir Dar City? Unless this is known it will be impossible to estimate the prevalence of depression among older adults in Bahir Dar City that can be generalized to the whole population of older adults in Bahir Dar City. 

Response: Thank you. We mentioned the total number of the older adult population. The study reported here is to investigate the prevalence of depression among the age group of 60 years and above, as measured by GDS-15, among 11,034 (5003 male and 6031 females) older adults in a city.

Line 119 Which information was taken from Bahir Dar City Municipality?

Response: For example The total population, the number of sub-cities, health care services including health centers is provided in the Bahir Dar city administration.

Line 124 All old age residents?? Consider correcting this statement.

Response: We corrected it as “All older adults who live in Bahir Dar city administration.”

Line 126 All old age population?? Same as in Line 124

Response We also corrected it as “Older adults aged 60 years and above who were available in Bahir Dar city administration during the study period.” See in the document.

Line 128 Consider correcting this sentence.

Response: We amended it accordingly.

Line 133, 134 consider reframing the statements. Was the sample size estimated using Epi info 7 Stat Calc software or single population proportion formula?

Response: Corrected it as “The sample size was calculated using the single population proportion formula with the assumption of a prevalence (P) of depression of 47.5 % from a previous study with a confidence limit of 5%. As a result, n=384, with no requirement for a correction factor because the population size is more than ten thousand. The ultimate sample size was 423 after adding a 10% non-response rate”. See the main document.

Line 141 Which population size are you referring to here; the total population or the population of older adults aged 60 and over?

Response: Thank you for critical view. We put the total older adult population which was 11,034. See the main document.

Line 142 Clarify how the final sample size was distributed proportionally among the eight sub-cities.

Response: We revised it as “Among these, the age group of 60 years and above is estimated to be 11,034 (5003 male, and 6031 females). Those older adults are Shimbit (1670), Tana (1043), Fasilo (1200), Sefene selam (287), Gishabay (522), Shum ambo (417), Belay Zeleke (1591), and Ginbot-20 (4304)”. From the above total older adults we allocated proportionally and from this, Shimbit (64), Tana (40), Fasilo (46), Sefene Selam (11), Gish Abay (20), Shum Abbo (16), Belay Zeleke (61), and Gimbot Haya (165) were included in this study.

Line 144 sample frame? od ages? Correct.

Response: Thank you, corrected.

Line 151 Limitations of MMSE should be addressed

Response: Thank you! We added some limitations of MMSE in the section on limitations

For example: “The generalizability of the study might be limited for those who had formal education since the tool (MMSE) is adapted based on the educational level in this setup.”

Line 156 Has this instrument been validated in Ethiopia?

Response: The revised Multidimensional Scale of Perceived Social Support (MSPSS) was employed to evaluate social support. It was developed by Zimet et al.,1988. And identifies three sources of support (friends, family, and significant others). The tool was widely used among old age, pregnant mothers, and other populations and has been validated in one African country (Cronbach's alpha = 0.916) (Stewart et al., 2014). In the Ethiopian context, the tool has not been validated yet and the Cronbach's alpha was 0.81 in the current study.

Line 170 The limitations of assessing income using this method should be addressed.

Response: It is simplistic and does not reflect indebtedness, health, education, housing, or access to public services. Also, it does not always accurately represent the different costs of subsistence from country to country. Difficulty to assess each item found in the household, despite this limitation many developing country studies use this assessment way. It's better to assess the wealth index using PCA for future researchers. W put it in the limitation section. Thank you for your very supportive comment.

Line 176 Has GDS been validated in Ethiopia? Which version of GDS was used? Validated or original version?

Response: Geriatric Depression Scale item 15 (GDS-15) was used to assess the presence of depression among older adults. Even if the tool hasn’t been validated in Ethiopia we use extensively validated in low and middle-income countries such as India, Nepal, other Asian and African countries with a sensitivity of 92% and specificity of 89%. Also, various studies conducted in Ethiopia use this instrument to assess depression. The validity and reliability of the tool have been supported through both clinical practice and research. In addition, we conducted a pretest and its internal consistency (Cronbach alpha) in this study was 0.86. Finally, we do have a plan to conduct a validation study for the future.

Line 182 What about the older adults with education below fifth grade, and those who could neither read nor write? Were they excluded from the study? The limitations of using MMSE to assess cognition should be appreciated. How was dementia distinguished from depression in this study?

Response: Yes! we excluded older adults with education below fifth grade and we revised in the exclusion criteria accordingly. One important limitation of MMSE is that it cannot be administered to illiterate subjects as 2 of its items involve reading and writing. Also, the limitation is the inclusion of a task requiring paper and pencil (copying a drawing). Furthermore, difficulty in identifying mild cognitive impairment. Whereas, Depression vs dementia. We put this as a limitation because of the nature of the cross-sectional study design, it is impossible to establish cause and effect linkages.

Lines 200,201.202 , 204 Consider correcting these statements. What is MSc?

Response: MSc means in this context, Masters of Sciences degree holder in mental health. And we amended it as “The data were collected from study participants by face-to-face interviews from house to house. The questionnaire was prepared in English and then translated into the local language (which is Amharic) by a language translator and translated back to English to ensure its understandability and consistency before the actual data collection. The training was given for supervisor and data collectors by the principal investigator for two days duration on the methods of data collection and the detail of the questionnaire. Data were collected by four psychiatric nurses who currently work in health centers and was supervised by Masters of Sciences degree holder in mental health”.

Lines 202, 203,204 Was the GDS translated into the local language(Amharic)?

Response: The questionnaires were prepared in English and then translate into the local language Amharic, and back-translated to English by an independent person to ensure its understandability and consistency.

Lines s 207, 2008,209. Consider correcting these statements.

Response: We revised it as recommended.

Line 222 How was the confidence interval of the prevalence calculated?

Response: Using SPSS we perform bootstrapping, In short, Go to Analyze-descriptive statistics- frequencies-drag the variable (DV) to the variable box – perform bootstrap with 95% CI. Finally, the results show the prevalence in yes/no with 95% CI (upper and lower limit). In this study, the overall prevalence of depression among elderly people in Bahir Dar city was with CI 272 57.9 % (95% CI: 53.2,62.6)

Line 235 How about the older adults with a level of education below the fifth grade and those who could neither read nor write? Were they excluded from the study?

Response: Thank you for your critical view. They were excluded from the study. And we corrected the exclusion criteria. Again, thank you, dear reviewer!

Line 265 correct this statement.

Response: We corrected it as “ Low parental relationships support, low friend social support, and low other social support were found in 55 (13.1%), 56 (13.2%), and 57 (13.5%) of the participants, respectively”.

Line 271 shows how the prevalence of depression was calculated. How was the confidence interval calculated?

Response: We explained in the above response. The prevalence of depression among older adults is calculated by measuring the presence of depression in a sample of the population selected randomly, then dividing the number of older adults that have depression by the number of people in whom it was measured. Prevalence is often expressed as a percentage.

Simply,

 Prevalence= (Total cases)/(Total population) x 100, P=245/423 x 100 = 57.9

Or, Using software analysis (SPSS), we did a simple descriptive analysis for prevalence. Additionally, see the SPSS output below (see below tables).

-Descriptive analysis for depression prevalence 

Depression prevalence

 Frequency Percent Valid Percent Cumulative Percent

Valid NO 178 42.1 42.1 42.1

 YES 245 57.9 57.9 100.0

 Total 423 100.0 100.0 

CI FOR DEPRESSION

Depression catagory

 Frequency Percent Valid Percent Cumulative Percent Bootstrap for Percenta

 Bias Std. Error 95% Confidence Interval

 Lower Upper

Valid NO 178 42.1 42.1 42.1 .0 2.4 37.4 46.8

 YES 245 57.9 57.9 100.0 .0 2.4 53.2 62.6

 Total 423 100.0 100.0 .0 .0 100.0 100.0

a. Unless otherwise noted, bootstrap results are based on 10000 bootstrap samples

Lines 283 to 292 Consider correcting the statements with 3.44 times, 2.78 times etc.

Response: We corrected it as needed.

Line 313 to 328 Consider summarizing this paragraph.

Response: We revised and rewite it again as you recommend. 

Line 334 Low education may limit the development of therapies to alleviate the disease burden of depression. Can older adults develop therapies for depression? Please consider correcting this statement.

Response: Revised.

Lines 339 to 342 Consider correcting the statements.

Response: We revised it accordingly. We rewrite it as “This is the finding that low-income people have a harder time getting healthy services and care, which has been associated with higher levels of depression. McCall and colleagues' findings in the United States supported prior studies that connected low income to a higher prevalence of depression.”

Line 343 correct 3.54 times

Response: We think it is better to write it as “Older adults who had cognitive impairments were 3.54 times more likely to develop depression compared with their counterparts.”

Line 349 limitations of MMSE in assessing cognitive impairment should be addressed.

Response: We tried to incorporate the limitation in the limitations section.

Line 352 populations who had family history of mental illness? Consider correcting this.

Response: We replaced it with “older adults”.

Line s 376 to 382 Correct the statements .eg we recommend that clinicians regularly screening depressive symptoms using standard studies in the elderly

Response: We revised it accordingly. See the conclusion section in the main document.

Line 399 What about the older adults who were illiterate? Were they excluded from the study? Were there older adults who were unable to provide informed consent due to cognitive impairment? How many older adults had severe cognitive impairment per MMSE?

Response: We excluded older adults with education below fifth grade and we revised the exclusion criteria accordingly. Almost all those elder adults who had severe NCD using MMSE score 17, which is near to moderate. All older adults in this study provided informed consent and were answered questionnaires.

MMSE level

 Frequency Percent Valid Percent Cumulative Percent

Valid Mild 246 58.2 58.2 58.2

 moderate 169 40.0 40.0 98.1

 severe 8 1.9 1.9 100.0

 Total 423 100.0 100.0 

Line 404 Any time during the procedure?

Response: We revised it. 

Lines 404, 405 correct the statements.

Response: We corrected as needed. “ The information was not disseminated without the respondent's permission. The information provided by the participants was exclusively utilized for the study. Those older adults who reported depression were immediately referred to mental health facilities for further evaluation and management.”

Typos and Grammatical errors

There several typos and grammatical errors in this manuscript apart from those highlighted . These should to be corrected

Response: We tried to assess errors like incomplete sentences, grammatical and language errors from the title up to a discussion of the manuscript. In addition to that the name of the professional that edit our manuscript for language usage, spelling, and grammar service that provided: Mr. Demise Arega (BA, MA in TEEFL) from Wollo University at Department of language and literature 

Email: demisarega13@gmail.com

Reviewer 2

The authors have presented findings which adds to current literature on depression. Overall the methods chosen were appropriate and the results support the conclusions. However, the authors need to improve on the language of the manuscript.

Response: Thank you, dear reviewer, it helps to do more for the future. And we tried to assess errors like incomplete sentences, grammatical and language errors from the title up to a discussion of the manuscript. In addition to that the name of the professional that edit our manuscript for language usage, spelling, and grammar service that provided: Mr. Demise Arega (BA, MA in TEEFL) from Wollo University at Department of language and literature 

Email: demisarega13@gmail.com

Sincerely,

Tamrat Anbesaw, January/2022

---

## [Decision Letter · Decision Letter 1]

21 Jun 2022

PONE-D-21-38673R1Depression and associated factors among older adults in Bahir Dar city administration, Northwest Ethiopia, 2020: Cross-sectional studyPLOS ONE

Dear Dr.Tamrat,

Thank you for resubmitting your manuscript to PLOS ONE. After careful consideration, we feel that it has merit but does not fully meet PLOS ONE’s publication criteria as it currently stands. Therefore, we invite you to submit a revised version of the manuscript that addresses the points raised during the review process.

Please review your paper and make the manuscript clear and reader-friendly with following concerns wit hother reviewers comments:

1. Abstract results section line 34 CI 53.2, 62.6 should be consistent with other CIs

2. Make an abstract conclusion very specific based on your findings and provide pragmatic suggestions which are very vague now. Provide at least 5 appropriate keywords.

3. Line 74 writes in word one instead of 1 million.

4. In the methods and materials section starts with the research design followed by the study population (remove line 122) and merges the study population with inclusion and exclusion criteria, study sites, sample size, and sampling techniques,..

5. line 207 removes processing and keeps only data analysis

6. Line 219 writes down Ethical consideration

7. Line 245 Results

8. Line 247-258 presents very key findings in only % and refer to Table 1 for details.

9. Present mean/median age with SD or IQR

10. Review table 1 educational status variables and present constantly i.e 5-8th vs 9-12th

11. Table 1 presents the mean/median monthly income with SD/IQR

12. Line 261-267 review findings and present key findings with % and recommend Table 2 for the details.

13. Review the lines 271-273 as per above.

14. Line 285 headings and 290 should merge and start with prevalence instead of separate headings and also make consistent terminology of the figure headings older adults instead of elderly

15. Line 280 - 307 present only key findings

16. Line 307 correct error is showing not sure what is written.Please include the following items when submitting your revised manuscript:A rebuttal letter that responds to each point raised by the academic editor and reviewer(s). You should upload this letter as a separate file labeled 'Response to Reviewers'.A marked-up copy of your manuscript that highlights changes made to the original version. You should upload this as a separate file labeled 'Revised Manuscript with Track Changes'.An unmarked version of your revised paper without tracked changes. You should upload this as a separate file labeled 'Manuscript'.We look forward to receiving your revised manuscript.

Kind regards,

Sharada Prasad Wasti, Ph.D., MSc, MHCM, MA.

Academic Editor

PLOS ONE

Journal Requirements:

Additional Editor Comments:

Dear Authors,

Thank you for your thorough review and resubmission. Our reviewer has provided the following minor comments to readdress for this manuscript.

Main Issues not addressed by the authors:

1. Prevalence calculation.

The authors have stated that they calculated the prevalence of depression as follows:

Prevalence= (Total cases)/ (Total population) x 100, P=245/423 x 100 = 57.9%

This is prevalence of depression among the older adults screened for depression in Bahir Dar City not the prevalence of depression among older adults in Bahir Dar City.

The authors stated that the total population of older adults aged 60 years and above in Bahir Dar City is 11,034 (5003 male and 6031 females). The correct approach would be estimating the number of cases of depression in each sub-city, adding the cases and then diving by the total population ( 11,034).

2. Informed consent.

In their response to reviewers’ comments, the authors stated that almost all older adults had severe cognitive impairment. However in the manuscript only 42% had cognitive impairment. It is not clear if these older adults with severe cognitive impairment had capacity to provide a valid informed consent. This should be clarified.

3. Abstract

The conclusion does not reflect what the authors have studied. For example the study did not look at supportive therapy for treatment of depression in this population.

4. Other issues.

There are several grammatical errors in this manuscript. Eg lines 172-176. These should to be corrected.

Line ‘hard time’, use scholastic language

Line 5 and 6, empty space

Lines 11-19, empty space

Lines 47 -49, empty space

Lines 96-105 , empty space

Line 206, empty space

Lines 229-244 empty space

Lines 282- 284 empty space

Lines 308- 311 empty space

Lines 318, 395, 405, 438-440 empty space

References

Several references eg no 17, 29, 32 are incorrectly cited. This should be corrected.

Correct all above suggestions and submit for the further proceed it.

Reviewers' comments:

Reviewer's Responses to Questions

**Comments to the Author**

1. If the authors have adequately addressed your comments raised in a previous round of review and you feel that this manuscript is now acceptable for publication, you may indicate that here to bypass the “Comments to the Author” section, enter your conflict of interest statement in the “Confidential to Editor” section, and submit your "Accept" recommendation.

Reviewer #1: (No Response)

2. Is the manuscript technically sound, and do the data support the conclusions?

Reviewer #1: Partly

3. Has the statistical analysis been performed appropriately and rigorously? 

Reviewer #1: No

4. Have the authors made all data underlying the findings in their manuscript fully available?

Reviewer #1: Yes

5. Is the manuscript presented in an intelligible fashion and written in standard English?

Reviewer #1: No

6. Review Comments to the Author

Reviewer #1: Main Issues not addressed by the authors

1. Prevalence calculation.

The authors have stated that they calculated the prevalence of depression as follows:

Prevalence= (Total cases)/ (Total population) x 100, P=245/423 x 100 = 57.9%

This is prevalence of depression among the older adults screened for depression in Bahir Dar City not the prevalence of depression among older adults in Bahir Dar City.

The authors stated that the total population of older adults aged 60 years and above in Bahir Dar City is 11,034 (5003 male and 6031 females). The correct approach would be estimating the number of cases of depression in each sub-city, adding the cases and then diving by the total population ( 11,034).

2. Informed consent.

In their response to reviewers’ comments, the authors stated that almost all older adults had severe cognitive impairment. However in the manuscript only 42% had cognitive impairment. It is not clear if these older adults with severe cognitive impairment had capacity to provide a valid informed consent. This should be clarified.

3. Abstract

The conclusion does not reflect what the authors have studied. For example the study did not look at supportive therapy for treatment of depression in this population.

4. Other issues.

There are several grammatical errors in this manuscript. Eg lines 172-176. These should to be corrected.

Line ‘hard time’, use scholastic language

Line 5 and 6, empty space

Lines 11-19, empty space

Lines 47 -49, empty space

Lines 96-105 , empty space

Line 206, empty space

Lines 229-244 empty space

Lines 282- 284 empty space

Lines 308- 311 empty space

Lines 318, 395, 405, 438-440 empty space

References

Several references eg no 17, 29, 32 are incorrectly cited. This should be corrected.

7. PLOS authors have the option to publish the peer review history of their article (what does this mean?). If published, this will include your full peer review and any attached files.

Reviewer #1: **Yes: **Dr Damas Andrea Mlaki

---

## [Author Response · Author response to Decision Letter 1]

22 Jun 2022

Response to editor and reviewers

Please review your paper and make the manuscript clear and reader-friendly with following concerns with other reviewers comments:

1. Abstract results section line 34 CI 53.2, 62.6 should be consistent with other Cis

Response: We corrected as recommended. 

2. Make an abstract conclusion very specific based on your findings and provide pragmatic suggestions which are very vague now. Provide at least 5 appropriate keywords.

Response: We revised it extensively. See the manuscript. Thank you dear editor to give achance to see the manuscript again.

3. Line 74 writes in word one instead of 1 million.

Response: Corrected.

4. In the methods and materials section starts with the research design followed by the study population (remove line 122) and merges the study population with inclusion and exclusion criteria, study sites, sample size, and sampling techniques,..

Response: We corrected it as per your recommendation.

5. line 207 removes processing and keeps only data analysis

Response: Thank you, we corrected as per a recommendation. 

6. Line 219 writes down Ethical consideration

Response: We revised it accordingly. 

7. Line 245 Results

Response: We corrected it.

8. Line 247-258 presents very key findings in only % and refer to Table 1 for details.

Response:We corrected it.

9. Present mean/median age with SD or IQR

Response: We present it as recommended. 

10. Review table 1 educational status variables and present constantly i.e 5-8th vs 9-12th

Response: Thank you, we corrected it.

11. Table 1 presents the mean/median monthly income with SD/IQR

Response: Thank you, we corrected it as per recommendation.

12. Line 261-267 review findings and present key findings with % and recommend Table 2 for the details.

Response: Thank you, we revised it. 

13. Review the lines 271-273 as per above.

Response: Again we thank you, and revised as you recommended. 

14. Line 285 headings and 290 should merge and start with prevalence instead of separate headings and also make consistent terminology of the figure headings older adults instead of elderly

Response: We revised it as per your recommendation. See the revised manuscript.

15. Line 280 - 307 present only key findings

Response: Thank you, dear editor, in this we tried to show the important points. If specific things to be revised we are welcome. 

16. Line 307 correct error is showing not sure what is written.

Response: In line 307, we found this value (AOR: 2.78, 95% CI: 1.74–4.46), which was written correctly in table 4. If any issues again we are welcome to revise it. 

 Response to reviewers

1. Prevalence calculation.

The authors have stated that they calculated the prevalence of depression as follows:

Prevalence= (Total cases)/ (Total population) x 100, P=245/423 x 100 = 57.9%

This is prevalence of depression among the older adults screened for depression in Bahir Dar City not the prevalence of depression among older adults in Bahir Dar City.

The authors stated that the total population of older adults aged 60 years and above in Bahir Dar City is 11,034 (5003 male and 6031 females). The correct approach would be estimating the number of cases of depression in each sub-city, adding the cases and then diving by the total population ( 11,034).

Response: Thank you dear reviewer. 

Prevalence is calculated as follows;

1. To estimate prevalence, researchers randomly select a sample (smaller group) from the entire population they want to describe.

2. For a representative sample, prevalence is the number of people in the sample with the characteristic of interest, divided by the total number of people in the sample.

-If we calculate by including the total population (11,034), we are saying we didn’t need a sample size calculation. Analysis should be conducted based on collected data or sample size included in the study (423). So we put it as it is.

2. Informed consent.

In their response to reviewers’ comments, the authors stated that almost all older adults had severe cognitive impairment. However, in the manuscript only 42% had cognitive impairment. It is not clear if these older adults with severe cognitive impairment had capacity to provide a valid informed consent. This should be clarified.

Response: Thank you, dear reviewer. Previously we respond only 8(1.9%) had severe cognitive impairment. Informed (written) consent was obtained from each study participant. The study participants were also provided with information about the objectives and expected outcomes of the study. Information obtained from individual participants was kept secure and confidential. We had given time to ask questions repeatedly for better thinking, understanding, and responding to the questions.

3. Abstract

The conclusion does not reflect what the authors have studied. For example, the study did not look at supportive therapy for the treatment of depression in this population.

Response: We revised it accordingly as per your recommendation. Again, thank you!

4. Other issues.

There are several grammatical errors in this manuscript. Eg lines 172-176. These should to be corrected.

Response: We revised it accordingly. See the main manuscript.

Line ‘hard time’, use scholastic language

Response: Dear reviewer, thank you for your recommendation to revise it. We replaced it with “more difficult”

Line 5 and 6, empty space

Lines 11-19, empty space

Lines 47 -49, empty space

Lines 96-105 , empty space

Line 206, empty space

Lines 229-244 empty space

Lines 282- 284 empty space

Lines 308- 311 empty space

Lines 318, 395, 405, 438-440 empty space

Response: We corrected all empty spaces. 

-References

Several references eg no 17, 29, 32 are incorrectly cited. This should be corrected.

Response: We corrected it as per recommendation.

Regards,

Tamrat Anbesaw( Corresponding Author)

---

## [Editor Report · Decision Letter 2]

25 Jul 2022

PONE-D-21-38673R2Depression and associated factors among older adults in Bahir Dar city administration, Northwest Ethiopia, 2020: Cross-sectional studyPLOS ONE

Dear Tamrat Anbesaw,

Thank you for your corrected version whch looks perfect. After careful consideration, we feel that it has merit but does not fully meet PLOS ONE’s publication criteria as it currently stands. Therefore, we invite you to submit a minor revised version of the manuscript that addresses the points raised during the review process. **Could you please remove line 94 and 95 Source Population and All older adults who live in Bahir Dar city administration which you have talked in line 83.**We look forward to receiving your revised manuscript for the final decision.

Kind regards,

Sharada Prasad Wasti, Ph.D.

Academic Editor

PLOS ONE

---

## [Author Response · Author response to Decision Letter 2]

27 Jul 2022

Dear Sharada Prasad Wasti, Ph.D., Academic Editor, thank you dear Editor for giving a chance to revise it again. We have seen the manuscript critically and revised as per recommendation.

---

## [Editor Report · Decision Letter 3]

8 Aug 2022

Depression and associated factors among older adults in Bahir Dar city administration, Northwest Ethiopia, 2020: Cross-sectional study

PONE-D-21-38673R3

Dear Tamrat Anbesaw,

We’re pleased to inform you that your manuscript has been judged scientifically suitable for publication and will be formally accepted for publication once it meets all outstanding technical requirements.

Kind regards,

Sharada Prasad Wasti, Ph.D., MSc, MHCM, MA.

Academic Editor

PLOS ONE
---

## [Editor Report · Acceptance letter]

11 Aug 2022

PONE-D-21-38673R3 

Depression and associated factors among older adults in Bahir Dar city administration, Northwest Ethiopia, 2020: Cross-sectional study 

Dear Dr. Anbesaw:

I'm pleased to inform you that your manuscript has been deemed suitable for publication in PLOS ONE. Congratulations! Your manuscript is now with our production department. 

Kind regards, 

on behalf of

Dr. Sharada Prasad Wasti 

Academic Editor

PLOS ONE